# Shared decision-making for children with medical complexity in community health services: a scoping review

Sonja Jacobs  ,[1] Nathan Davies,[2] Katherine L Butterick,[1] Jane L Oswell,[1] Konstantina Siapka,[1] Christina H Smith[3]

¹Community Children's Therapies, Barts Health NHS Trust, London, UK
²Research Department of Primary Care & Population Health, University College London, London, UK
³Division of Psychology and Language Sciences, University College London, London, UK

**Correspondence to**
Dr Christina H Smith; christina.smith@ucl.ac.uk

Mrs Sonja Jacobs; sonja.jacobs@nhs.net

## ABSTRACT

**Background** Children with medical complexity is an increasing population whose parents and healthcare providers face multiple decisions. Shared decision-making is a process where patients, their families and healthcare providers collaborate to make decisions based on clinical evidence and informed preferences of the family. Shared decision-making has benefits for the child, family and healthcare providers, including improved parental understanding of the child's difficulties, increased participation, improved coping skills and more efficient healthcare use. It is, however, poorly implemented.

**Aims and methods** A scoping review was conducted to explore shared decision-making for children with medical complexity in community health services, including how shared decision-making is defined in research, how it is implemented, including barriers and facilitators and recommendations for research. Six databases were systematically searched for papers published in English up to May 2022: Medline, CINAHL, EMBASE, PsycINFO, PubMed, Cochrane Database of Systematic Reviews and sources of grey literature. The review is reported according to the Preferred Reporting Items for Scoping Reviews.

**Results** Thirty sources met the inclusion criteria. Most factors can either be a facilitator or barrier to shared decision-making depending on the context. Two significant barriers to shared decision-making in this population include uncertainty about the child's diagnosis, prognosis, and treatment options and the presence of hierarchy and power imbalance during clinical encounters with healthcare providers. Further influencing factors include continuity of care, the availability of accurate, accessible, adequate, and balanced information and the interpersonal and communication skills of parents and healthcare providers.

**Conclusion** Uncertainty about diagnosis, prognosis and treatment outcomes for children with medical complexity are additional challenges to the known barriers and facilitators to shared decision-making in community health services. Effective implementation of shared decision-making requires advancement of the evidence base for children with medical complexity, reducing power imbalance in clinical encounters, improving continuity of care, and improving the availability and accessibility of information resources.

## WHAT IS ALREADY KNOWN ON THIS TOPIC

⇒ Shared decision-making is an evidence-based approach with known benefits to children with medical complexity, their families and the healthcare system, it is, however, poorly implemented.

## WHAT THIS STUDY ADDS

⇒ This study highlights the impact of uncertainty of diagnosis, prognosis and treatment outcomes for children with medical complexity on shared decision-making. It highlights how healthcare providers can improve the implementation of shared decision-making by addressing the power imbalance in clinical encounters, improving continuity of care, improving communication and interpersonal skills, and making information more accessible to parents from diverse backgrounds.

## HOW THIS STUDY MIGHT AFFECT RESEARCH, PRACTICE OR POLICY

⇒ This review can guide a research strategy in the field of shared decision-making for children with medical complexity in community health services support healthcare professionals to consider their influence on the decision-making process in everyday practice.

## INTRODUCTION

Children with medical complexity (CMC) have needs in four domains, namely (1) substantial family-identified healthcare and special educational needs, (2) one or more severe and potentially lifelong chronic conditions, (3) limitations to body structure and function, performance of activities and participation that may require technological assistance such as feeding tubes and (4) high projected healthcare use including the involvement of multiple subspecialties.[1] These children are increasing in number.[1 2] Parents and healthcare providers (HCPs) for CMC face multiple, complex decisions throughout their childhood including decisions about tube feeding, mechanical ventilation, medications and surgery.[1 3 4]



**Table 1** Eligibility criteria

| | Included | Excluded |
|---|---|---|
| **Population** | Parents and/or caregivers of children under 18 with medical complexity | Adult patients |
| | Children with specific medical diagnoses if they meet criteria for medical complexity | Children with behavioural, emotional or mental health conditions (eg, autism, depression, attention deficit hyperactivity disorder) |
| | Healthcare providers for children with medical complexity | |
| **Concept** | Shared decision-making in the paediatric clinical context | Decisions about vaccinations or public health issues |
| | | Pregnancy, perinatal or viability decision-making |
| | | Decision about participating in research |
| **Context** | Outpatients or tertiary care settings | Inpatient hospital settings |
| | Children's community health services | Primary care settings |
| | Contexts spanning multiple settings if they address paediatric community healthcare delivery | Settings outside healthcare (ie, education) |
| | | Universal health service |
| | | Prevention programmes |

Shared decision-making (SDM) is an evidence-based approach that is an essential part of patient-centred care.[5–7] It is a process where parents, as surrogate decision-makers for their child, and HCPs work in partnership to make decisions based on clinical evidence and family preferences.[7–10] This approach is supported by policy makers and regulatory bodies nationally and internationally.[5–10] The benefits for patients, families and HCPs include improved patient or carer knowledge and understanding, reduced decisional conflict, increased participation and engagement in care, improved coping skills, and efficient use of healthcare resources.[5 6 8 10] SDM is, however, poorly defined due to

**Table 2** Text words, index terms and subject heading identified for full search

| Key concept | Text words/index terms/MeSH terms (Medical Subject Headings)—combined using Boolean operators AND/OR |
|---|---|
| **Children** | child; child, preschool; adolescent; infant, extremely premature; infant; infant, newborn; paediatric |
| **Medical complexity** | medical complexity; special healthcare needs; disabilities; assistive technology; disabled children; developmental delay; chronic disease/th (therapy); nervous system diseases/th (therapy); medical fragility |
| **Shared decision-making** | Parental decision-making; shared decision-making; parent perspective; decision-making; patient participation; family-centred care; patient-centred care; professional-family relationship; parental discretion; bioethical issues |

the interpretive nature of what is meant by 'shared',[6 11] with fundamental differences in how patients, carers and HCPs understand the purpose of and their role in SDM.[6 8 12–14]

The difference between parental and HCP approaches to decision-making often result in poor implementation of SDM. HCPs base their decisions on clinical and empirical evidence,[8] which is often lacking for CMC.[1 15] Parents consider the social, emotional and psychological impact of decisions on their child, their family and cultural and religious beliefs in addition to potential clinical outcomes.[8 11 13] Parental decisions about what is 'good enough' for their child with medical complexity are often more intensely scrutinised by HCPs than for non-medically complex children, with a lack of awareness or importance given to the impact of decisions on the family.[8 11 13–16] Clinical uncertainty combined with complex family dynamics require HCPs to swap traditional hierarchical and paternalistic approaches to decision-making, where decisions are made based on clinical information and empirical evidence,[5 8 11 12 15 17] for an approach that allows parental collaboration and discretion in decision-making.[11 15]

The personal and healthcare cost of poor implementation of SDM is amplified in the CMC population due to their significant healthcare use. Understanding factors impacting SDM for CMC will help to improve medical and developmental outcomes, quality of life of children and families and effective use of healthcare resources.[5 8 10]

This scoping review aimed to explore the landscape of SDM for CMC in community health services.

The objectives for this review were to:
1. Explore how SDM is defined in research.
2. Understand to what extent SDM is implemented for CMC in community health services.

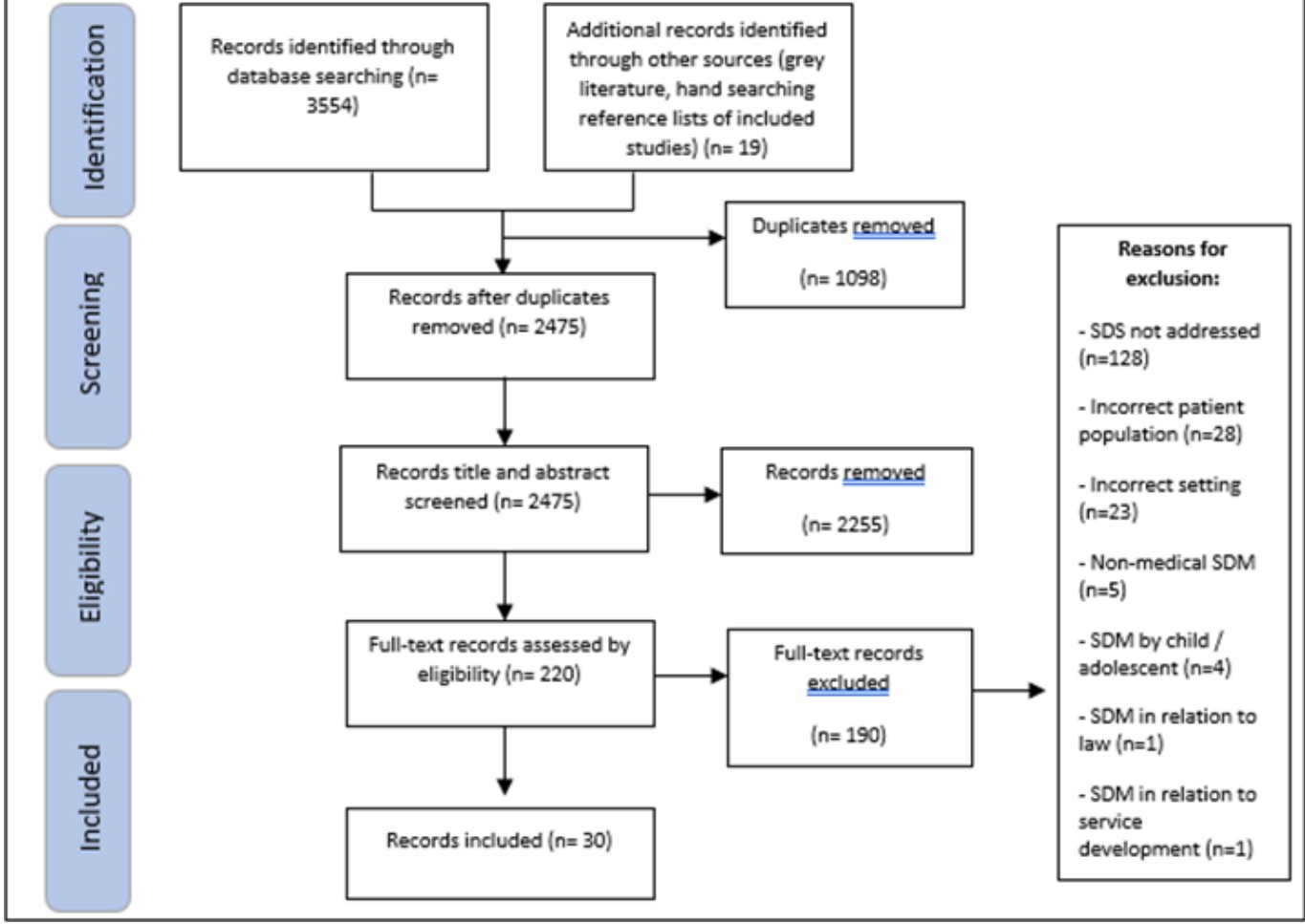

**Figure 1** PRISMA flow chart. PRISMA, Preferred Reporting Items for Systematic Reviews and Meta-Analyses; SDM, shared decision-making.

3. Consider the differences in SDM between ethnic groups.
4. Identify the barriers and facilitators to SDM for CMC.
5. Provide recommendations for future research.

## METHODS

A scoping review was conducted following the Joanna Brigs Institute (JBI) manual for evidence synthesis[18–20] and was reported according to the Preferred Reporting Items for Systematic Reviews and Meta-Analyses (PRISMA) extension for scoping reviews.[21]

### Protocol and registration

A scoping review protocol was registered on Open Science Framework on 19 May 2022.[16]

### Eligibility criteria

Eligibility criteria are outlined in table 1. SDM included any process involving parents or caregivers in medical decision-making with HCPs[10] and included family-centred practices. Sources reporting on multiple populations or settings were included if results were reported separately or if at least 50% of the results related to the eligible population or setting. Primary research using any methodology, secondary research including systematic reviews, literature, and scoping reviews and editorial or opinion pieces were included.

### Information sources

The search included literature published from 1982 when SDM was first mentioned in scientific literature.[9] Only articles published in English were included due to the time and cost of transcription. A three-step search strategy was followed.[18] An initial search of Medline and CINAHL identified text words and index terms to develop a full search strategy (table 2). This search strategy was reviewed by a librarian using the Peer Review of Electronic Search Strategies 2015 guideline[22] and was used to search databases including Medline, CINAHL, EMBASE, AMED, PsycINFO, PubMed, Cochrane Database of Systematic Reviews and sources of grey literature including Open Grey, NICE guidelines and CanChild website. The reference lists of included sources were screened for additional sources. The final search was completed on the 26 May 2022.

### Selection of evidence

Identified sources were uploaded to Covidence systematic review software and duplicates removed. Titles

**Table 3** Description of sources—qualitative studies (ordered from earliest publication date)

| Lead author | Title | Country | Date | Aim |
|---|---|---|---|---|
| Brotherson[25] | Quality of life issues for families who make the decision to use a feeding tube for their child with disabilities | USA | 1995 | To explore the issues families face in deciding whether to place a feeding tube |
| Katz[26] | A cultural interpretation of early intervention teams and the IFSP: parent and professional perceptions of roles and responsibilities | USA | 1995 | To understand how members of an early intervention team involve families in developing of individual family service plans (IFSP). |
| Blue-Banning[27] | Dimensions of family and professional partnerships: constructive guidelines for collaboration | USA | 2004 | To identify indicators of professional behaviour indicative of collaborative partnerships. |
| Brotherton[28 29] | Mothers' process of decision-making for gastrostomy placement | UK | 2012 | To explore mothers' constructions of decision-making in gastrostomy feeding |
| Stille[29] | Parent partnerships in communication and decision-making about subspecialty referrals for children with special needs | Canada | 2013 | To describe factors influencing parent–clinician partnerships in SDM when children with special healthcare needs are referred to subspecialists. |
| Zaal-Schuller[30] | How parents and physicians experience end-of-life decision-making for children with profound intellectual and multiple disabilities | The Netherlands | 2016 | To compare experiences of parents and physicians involved in the end-of-life decision process |
| Buchanan[31] | What makes difficult decisions so difficult?: An activity theory analysis of decision-making for physicians treating children with medical complexity | Canada | 2020 | To first understand the complexity of the activity of decision-making |
| Lin[32] | Parent perspectives in SDM for CMC | USA | 2020 | To identify components of SDM unique to the care of CMC from the perspective of parents. |
| Jabre[33] | Parent perspectives on facilitating decision-making around paediatric home ventilation | USA | 2021 | To understand parent perspectives about how clinicians can better facilitate decision-making around home ventilation |
| Reeder[34] | Becoming an empowered parent. How do parents successfully take up their role as a collaborative partner in their child's specialist care? | UK | 2021 | To explore the important themes of dis/empowerment and the influence of the therapeutic relationship |
| Buchanan[4] | Decision-making for parents of children with medical complexities: activity theory analysis | Canada | 2022 | To explore decision-making of parents of CMC as an activity within the context of a process shared between clinician and parent |

CMC, children with medical complexity; SDM, shared decision-making.

and abstracts were screened against criteria by the first author and 33% was screened by second authors. Full texts of potentially relevant sources were assessed against inclusion criteria by the first author and 33% by second authors. Reasons for exclusion of sources at full text were recorded. Disagreements between the reviewers were resolved through discussion and a third reviewer if needed. The result of the search is outlined in a PRISMA flow chart (figure 1).

**Data charting process**

The JBI source of evidence template[18] was modified for extraction of details about the author, publication year, country, participants, aim, context, study methods and findings relevant to the review questions. The first author completed data extraction and 10% were checked for consistency by second authors. Discrepancy in extraction were resolved through discussion. The extraction tool

**Table 4** Description of sources—quantitative and mixed-methods studies (ordered from earliest publication date)

| Lead author | Title | Country | Date | Aim |
|---|---|---|---|---|
| Guerriere[39] | Mothers' decisions about gastrostomy tube insertion in children: factors contributing to uncertainty | Canada | 2003 | To explore mothers' perceptions of decision uncertainty. |
| Denboba[35] | Achieving family and provider partnerships for children with special healthcare needs | USA | 2006 | To assess whether families feel like partners in decision-making by their doctors |
| Pickering[40] | Disabled children's services: how do we measure family-centred care? | UK | 2010 | To evaluate staff and parental views of family-centred care in organisations providing services to young disabled children in Wales |
| Smalley[36] | Family perceptions of shared decision-making with healthcare providers: results of the National Survey of Children With Special Healthcare Needs, 2009–2010 | USA | 2014 | To use data from a national survey to determine families' perceptions of SDM and determine the sociodemographic correlates |
| Lin[3] | Shared Decision-Making among Children with Medical Complexity: Results from a Population-Based Survey | Canada | 2018 | To compare the rates of SDM reported by parents of CMC with the rates of SDM reported by parents of non-complex children with special healthcare needs |
| Jolles[38] | Shared decision-making and parental experiences with health services to meet their child's special healthcare needs: Racial and ethnic disparities | USA | 2018 | To test the relationship between SDM and parental report of frustration with efforts to get services for their child and to assess SDM's influence on minority parents' service experiences |
| An[37] | Effects of a Collaborative Intervention Process on Parent-Therapist Interaction: A Randomised Controlled Trial | South Korea | 2019 | To determine whether collaborative intervention impacted interactions between parents of children with physical disabilities and physical therapists |

CMC, children with medical complexity; SDM, shared decision-making.

was iterative and was updated as the researchers became more familiar with the evidence.

## Synthesis of results

Data were analysed by quantifying text, conducting basic qualitative content analysis and frequency counts.[23] Barriers and facilitators were ordered according to themes and mapped onto the ecological model of health behaviour (EMHB).[23 24] The EMHB emphasises the multiple layers of influence on healthcare behaviour and can guide the development of interventions by ensuring consideration of all factors impacting implementation.[24] Four ecological levels were used (1) individual level including factors related to the child's needs, (2) family level relating to knowledge, attitudes and skills of parents, (3) interpersonal level focused on interactions between HCPs and parents, and (4) organisational level considering institutional and HCP practices.

## Patient and public involvement statement

No patients were involved in conducting this scoping review.

## RESULTS
### Description of included sources

Thirty articles were included in this review, 18 were primary studies, 11 used qualitative[4 25–34] (table 3), 5 quantitative[3 35–38] and 2 mixed methods[39 40] (table 4). Eight articles were theoretical or opinion pieces[41–48] (table 5) and four literature reviews[49–52] (table 6). Twelve of the primary studies included parent participants, one included HCPs and five included parents and HCPs. Seventeen sources originated in the USA, seven in Canada, four in the UK, two in the Netherlands and one in South Korea. Research interest in SDM for CMC is increasing with 23 articles published in the last 10 years of which 15 were published in the last 5 years.

### Definition of SDM for CMC

Eighteen articles defined or described SDM (figure 2). A collaborative approach and equal partnership between parents and HCPs were most frequently noted as key elements of SDM. Most other elements offered guidance on how to achieve this partnership. Three sources

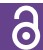

**Table 5** Description of sources—reviews (ordered from earliest publication date)

| Lead author | Title | Country | Date | Aim |
|---|---|---|---|---|
| Kruijsen-Terpstra[49] | Parents' experiences with physical and occupational therapy for their young child with cerebral palsy: a mixed studies review | The Netherlands | 2013 | To review literature on the experiences of parents of children with cerebral palsy with the physical and/or occupational therapy of their child. |
| Popejoy[50] | Decision-making and future planning for children with life-limiting conditions: a qualitative systematic review and thematic synthesis | UK | 2017 | To synthesise findings from qualitative research about decision-making and future planning for children with life-limiting conditions. |
| Jonas[51] | Parental Decision-Making for Children With Medical Complexity: An Integrated Literature Review | USA | 2022 | To consolidate existing literature on parental experience of medical decision-making for CMC. |
| LeGrow[52] | Relational Aspects of Parent and Home Healthcare Provider Care Practices for Children With Complex Care Needs Receiving Healthcare Services in the Home: A Narrative Review | Canada | 2022 | To review literature on relational aspects of parent and home healthcare provider care practices for children with complex healthcare needs. |

CMC, children with medical complexity.

referenced the impact of uncertainty of prognosis and treatment outcomes for CMC on this collaborative process.

**Implementation of SDM for CMC in community health services**
Two qualitative[26 34] and two quantitative[3 36] studies reported on the implementation of SDM with CMC in

**Table 6** Description of sources—theoretical or opinion (ordered from earliest publication date)

| Lead author | Title | Country | Date | Aim |
|---|---|---|---|---|
| Bazyk[41] | Changes in Attitudes and Beliefs Regarding Parent Participation and Home Programmes: An Update | USA | 1989 | To discuss traditional and current attitudes and practices regarding parent participation. |
| Arvedson[41] | Ethical and legal challenges in feeding and swallowing intervention for infants and children | USA | 2007 | To outline current state of evidence-based decision-making with feeding and swallowing. |
| An[43] | Family-professional collaboration in paediatric rehabilitation: a practice model. | USA | 2014 | To describe a practice model of family-professional collaboration for paediatric rehabilitation. |
| Austin[44] | Improving Partnerships to Make Family-Centred Care Work for Children with Special Healthcare Needs. | USA | 2014 | Explaining the importance of partnership working from a parental perspective |
| Adams[45] | Shared Decision-Making and Children with Disabilities: Pathways to Consensus. | USA | 2017 | To provide a basis for a systematic approach to implementation of SDM. |
| Madrigal[46] | Supporting Family Decision-making for a Child Who Is Seriously Ill: Creating Synchrony and Connection | USA | 2018 | To discuss the process of supporting families facing chronic and serious illness during decision-making. |
| Mahant[47] | Decision-making around gastrostomy tube feeding in children with neurologic impairment: Engaging effectively with families | Canada | 2018 | To review evidence and conceptual frameworks and provide recommendations to support decisions about gastrostomies. |
| Lee[48] | Decision-Making for Children with Medical Complexity: The Role of the Primary Care Paediatrician. | USA | 2020 | Discussion of influences on decision-making from a paediatrician's perspective. |

SDM, shared decision-making.

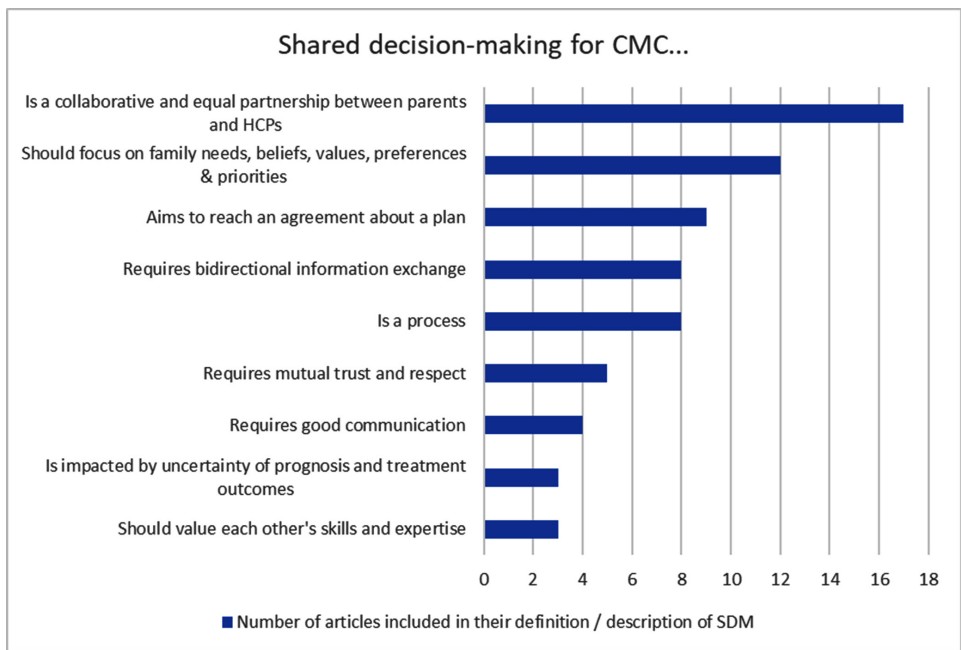

**Figure 2** Elements of shared decision-making (SDM) for children with medical complexity. CMC, children with medical complexity; SDM, shared decision-making.

community settings. The two qualitative studies were conducted 26 years apart (ethnographic study in 1995[26] and interview-based study in 2021[34]), in both parents perceived a power imbalance between them and HCPs with decision-making situated with HCPs. Two quantitative studies analysed the same dataset from a national survey in the USA[3 36] and found that although 85% of parents felt like partners with their child's doctor, there was a negative association with minority ethnic and low socioeconomic status[36] and children with greater complexity.[3] This was attributed to multidisciplinary support needs, frequent hospital admissions, clinical uncertainty and social difficulties often experienced by parents of CMC.

### Differences in SDM in different communities
Eight of the 18 primary studies reported on participant ethnicity but only three USA studies reported on differences in SDM among participants from a black, Hispanic and white background.[35 36 38] These studies found that families from minority ethnic backgrounds, those with lower educational backgrounds and lower income levels experienced less coordinated care and less SDM.

### Barriers and facilitators to SDM for CMC
Twenty-eight articles mentioned at least one barrier or facilitator to SDM. The most cited barriers related to clinical uncertainty of CMC, power imbalance between parents and HCPs and the lack of continuity of care (table 7).

The most cited facilitators to SDM for CMC included sharing of accessible, adequate, accurate and balanced information about all treatment options including knowing about uncertainty. Several facilitators related to the way HCPs viewed and engaged parents as active team members, service accessibility and attributes relating to the family (table 8).

### Research recommendations
Nineteen articles concluded with research recommendations, most related to discovering how to involve families and develop collaborative relationships, particularly families from diverse backgrounds (table 9).

### DISCUSSION
This scoping review explored the landscape of SDM for CMC in community health services. SDM is important in this population due to the complex long-term nature of their health conditions and high healthcare use. Like previous reviews,[6 10] this review found no unifying definition for SDM in the literature. All sources highlighted the importance of SDM; however, few studies explored the effectiveness of SDM for CMC, especially in community health settings. This might in part be due to the varying nature of service delivery models in different countries. The lack of implementation research is a shortcoming in paediatric research generally[10] with evidence mainly pertaining to adult care.[5 17] Limited research exists about SDM for CMC who are from a minority ethnic or disadvantaged background. Three studies showed poorer implementation in black and Hispanic communities in the USA.[35 36 38] This is congruent with evidence from a systematic review that included studies from 15 countries, showing that adults from minority ethnic or disadvantaged backgrounds experience more barriers to SDM.[53] Research shows that SDM interventions can significantly improve the outcomes for disadvantaged adult patients, including increased knowledge and participation in

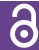

**Table 7** Barriers to SDM for CMC mapped to the ecological model

| | |
|---|---|
| **Individual (child) level—theme: uncertainty** | |
| Uncertainty about diagnosis or clinical management options | 3 28 29 31 32 36 45 47 48 50 |
| Lack of evidence and uncertain illness trajectories | 3 25 31 32 45 47 50 |
| Limited or conflicting information | 25 29 30 39 |
| Uncertainty about child's comfort and quality if life | 36 50 |
| **Family level** | |
| Language barriers | 29 36 38 40 45 |
| Poor general and health literacy | 25 28 45 50 |
| Lack of parental understanding of child's diagnosis and prognosis | 29 40 45 |
| Lack of trust in HCPs | 27 34 50 |
| Parents not feeling heard | 34 44 50 |
| Poverty, Black or minority ethnic background | 35 36 |
| Parental physical and emotional exhaustion and strong emotions | 32 50 |
| **Interpersonal level—theme: power imbalance** | |
| Hierarchy and power imbalance, coercive conversations by HCP, failure to explain options fully or withholding information and labelling parents as non-compliant if they disagree with HCPs | 3 4 26 28–30 32 34 36 44 45 48 50 |
| Using medical jargon and providing too detailed information | 4 27 33 40 43 45 51 |
| HCPs not valuing parental opinion and experience | 30 32 34 45 |
| **Oraganisational level—theme: lack of continuity of care** | |
| Involvement of multiple subspecialties, lack of continuity of care | 3 4 32 40 44 45 48 50 |
| Healthcare systems that dictate treatment options | 4 31 32 50 |
| Lack of time in consultations | 3 43 45 |
| Lack of professional understanding and valuing of SDM | 31 45 |
| Limited access to the right person to answer questions | 4 50 |

HCPs, healthcare providers; SDM, shared decision-making.

decision-making.[54] It is, therefore, not surprising that exploring SDM in diverse communities was one of the most cited research recommendations.

Many influencing factors could either be barriers or facilitators to SDM depending on the context.[5 7 53] Most factors identified in this review are situated in the organisational level where HCPs can influence change. This include organisational and personal understanding of and commitment to SDM, interpersonal skills that build trust and respect, active listening, cultural sensitivity, empowerment of families to be active team members and to share their values, opinions and fears, continuity of care, access to the right HCPs and enough time in consultations. Previous studies found that the top patient-reported barriers to SDM include disorganised healthcare systems and the quality of interactions with HCPs,[53] whereas the main barriers identified by HCPs are lack of time and motivation to pursue SDM and a perception that patients do not want to engage in decision-making.[57] There is, however, an awareness among HCPs that good communication and coordination of care can improve SDM.[2] To improve implementation of SDM it should be viewed as a culture within organisations and a way of interacting with every service user rather than another clinical tool.[2 10]

Another significant patient-identified barrier is hierarchy and power imbalance still prevalent in traditional approaches, where HCPs are the main decision-makers.[8] Even when espousing SDM, HCPs often provide biased information in order to achieve a specific decisional outcome, for instance by only providing information about their preferred treatment option.[7] This review found that HCPs often fail to explain all available options, withhold information and use jargon. In some instances, HCPs provide too much and too detailed information, also negatively impacting the decision-making process. Effective information exchange is an important step in the decision-making process[55] and can either decrease or increase the power balance in the relationship.

This power imbalance is compounded by clinical information being held by HCPs who can influence the accuracy, clarity, tailoring and sharing of that information.[7 13 53] This review identified accessible, adequate, accurate and balanced information as one of the most significant contributors to successful SDM and conversely, the lack of evidence and information as a barrier to SDM. Providing information that outlines options, risks and uncertainties can improve SDM[7 8] and if presented in an

**Table 8** Facilitators to SDM for CMC

| Individual (child) level | |
| --- | --- |
| Valuing the personhood of the child | 27 33 51 |
| **Family level** | |
| Parental comfort with decision-making | 28 32 38 50 52 |
| Knowledge of the healthcare system | 4 50 51 |
| Parental educational level | 31 50 |
| Parental understanding of the child's diagnosis and prognosis | 34 50 |
| **Interpersonal level** | |
| Mutual trust and respect | 4 27 29 35 46 48 49 |
| Actively empowering families to express their opinions, fears and hopes | 27 28 40 46 47 49 |
| Reciprocal good interpersonal skills | 27 30 31 40 50 |
| Reciprocal active listening | 27 34 35 40 46 |
| Regarding parents as experts on their child | 29 40 49–51 |
| Having shared goals | 27 44 46 48 |
| Sensitivity to cultural differences | 27 33 40 50 |
| Showing dedication to the family | 27 33 51 |
| Professional awareness of parental decision-making preferences | 46 47 |
| Respecting family decisions | 26 31 |
| **Organisational level—theme: information and access** | |
| Having accessible, sufficient, accurate and balanced information about all treatment options including knowing about uncertainty | 4 25–27 30 32 33 46 47 51 |
| Access to peer-to-peer support | 25 29 33 36 38 47 50 |
| Having sufficient time to consider information and knowing the time-horizon for decision-making | 30 32 33 47 50 51 |
| Access to information from non-professional sources such as social media | 31 32 39 47 50 |
| Including parents as members of the team | 27 36 40 |
| Continuity of care | 33 40 51 |
| Access to interpreters if needed | 29 40 |
| Access to the right healthcare professionals to answer questions | 4 29 |
| HCP seeking advice from the wider team in the face of uncertainty | 31 51 |
| Having access to written information | 29 30 |

CMC, children with medical complexity; HCP, healthcare provider; SDM, shared decision-making.

accessible and culturally sensitive format, can help overcome language and socioeconomic barriers such as poor literacy.[53]

A complicating factor in SDM for CMC is, however, the lack of clinical and empirical evidence and information due to the unique illness trajectory of CMC, leading to high levels of uncertainty. The presence of uncertainty was the most striking barrier to SDM for CMC found in this review and sets CMC apart from many other patient populations. A high level of uncertainty is not a typical feature of SDM for children[7] or adults[53] but has been cited in areas such as neonatal intensive care,[15] paediatric end-of-life care[15 56] and dementia end-of-life care.[55] The similarities with these clinical areas underscore the high-stress nature of decision-making for CMC.[13 14]

This review highlights the need for further research to increase the evidence base relating to diagnosis, prognosis and treatment options for CMC and to address the implementation of SDM for CMC, specifically focussing on families from diverse backgrounds who often experience less SDM.

### Strengths and limitations

This review contributes to the limited evidence base concerning SDM for CMC and highlights themes around uncertainty, power imbalance and information sharing on implementation of SDM. The risk of missing sources due to the number of synonymous terms for SDM[6] and CMC[1] used in the literature was mitigated by developing a comprehensive list of search terms and conducting a systematic search using a range of databases. Results were strengthened by

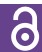

**Table 9** Research recommendations

| Discover how to involve families and develop collaborative relationships in SDM | 26 27 30 31 34 49 50 52 |
| --- | --- |
| Explore SDM within diverse communities, including diversity in family structure, culture and ethnicity, education level and healthcare setting | 30 32 36–38 49 51 52 |
| Explore family and healthcare professional's beliefs, perspectives and experiences of SDM | 30–32 43 51 52 |
| Develop guidelines for SDM | 27 36 46 50 |
| Evaluate the effect of professional training on SDM | 27 34 37 |
| Evaluate proposed models of SDM | 36 43 |
| Develop outcomes measures for SDM | 45 |
| Develop support technologies for SDM | 45 |
| Investigate information needs for effective SDM | 25 |
| SDM, shared decision-making. | |

having second reviewers at all screening stages. Healthcare organisation and configuration vary across and within countries. It can range from mainly hospital-based services to services delivered in various hospital and community settings, services can be offered free at the point of contact or require payment by insurers or service users. This variation could have impacted the identification of and comparability of studies. Most sources originated in the USA and Canada where barriers might be different to the UK and other parts of the world. The studies that included minority ethnic groups were conducted in the USA and do not represent the UK population. The review only included sources published in English, which might have resulted in the exclusion of potentially valuable papers.

## CONCLUSION

This scoping review revealed that uncertainty about diagnosis, prognosis and treatment outcomes for CMC has a significant impact on SDM, in addition to barriers and facilitators identified in other paediatric and adult populations, highlighting the need to advance the clinical evidence base for this population. Furthermore, many factors impacting SDM fall within the organisational level where HCPs can influence change, including pursuing a power balance and equal partnership, improving continuity of care and improving information resources to meet the needs of parents of CMC, including those from diverse backgrounds. Focusing on these factors can potentially improve medical and developmental outcomes, quality of life of children and families and more effective use of healthcare resources. This review can be used to guide a research strategy in the field of SDM for CMC in community health services.

**Contributors** SJ, ND and CHS planned the study. SJ is the guarantor and took the lead in performing the scoping literature search and data analysis and writing the manuscript. KB, JLO and KS were second reviewers for source selection and extraction and reviewed the manuscript. ND and CS supervised the study and reviewed the manuscript. All authors critically reviewed the final draft of the manuscript.

**Funding** SJ is an ICA Pre-doctoral Clinical Academic Fellow supported by Health Education England and the National Institute for Health Research, grant number: NIHR301944.

**Disclaimer** The views expressed in this publication are those of the authors and not necessarily those of the NHS, the National Institute for Health Research or the Department of Health and Social Care.

**Competing interests** No, there are no competing interests.

**Patient and public involvement** Patients and/or the public were not involved in the design, or conduct, or reporting, or dissemination plans of this research.

**Patient consent for publication** Not applicable.

**Provenance and peer review** Not commissioned; externally peer reviewed.

**Data availability statement** No data are available.

**ORCID iD**
Sonja Jacobs http://orcid.org/0000-0001-5021-0119

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
