## [Reviewer comments · BMJ Paediatrics Open]

ARTICLE DETAILS

TITLE (PROVISIONAL)	Shared decision-making for children with medical complexity in community health services – a scoping review
AUTHORS	Jacobs, Sonja; Davies, Nathan; Butterick, Katherine L; Oswell, Jane L; Siapka, Konstantina; Smith, Christina

VERSION 1 – REVIEW 1

REVIEWER	Reviewer Name: Dr. Diane Sellers Institution and Country: Chailey Clinical Services Speech and Language Therapy Beggars Wood Road, United Kingdom of Great Britain and Northern Ireland
REVIEW RETURNED	02-Feb-2023

GENERAL COMMENTS	This scoping review presents a really helpful summary of the current literature subdivided in to different categories (qual research, quant research and opinion piece). It is well organised, clearly presented and comprehensive in its outlook. The title suggests that the focus is on healthcare in a community setting although the search criteria and the papers you have included relate to both community and hospital based health care. E.g. Decisions about surgery are made with the surgeon; depending upon how services are organised, this may be with the family in acute hospital setting, hospital OP setting, or combined community clinic including surgeon providing outreach service. The configuration of decision making context and participants will influence the extent to which power imbalances are perceived and therefore changeable. I think you need to acknowledge that shared decision making in a community setting may not always be possible, especially when the key HCP involved in the decision is not present at the meeting. UK services are different across the country; North American Healthcare may be based strongly around hospitals with outreach in to the community. Differences in type of healthcare provided (insurance based, free healthcare with limitations, etc) will influence extent to which SDM can be considered. You may need to amend your title OR acknowledge the difficulty distinguishing between the two. Whilst I would support the notion that SDM is important in community healthcare, services will be configured differently to allow for different degrees of influence on critical healthcare decisions. The theme of "uncertainty" is a really key one and different health care professionals respond differently to "uncertainty" and this
--

	means decisions play out differently across different healthcare contexts. Because you have adopted headline QUAL review you have not explored themes underpinning each of the headings. Power hierarchies are rife within healthcare settings, especially in acute settings, with direct influence how decisions are made with families. You point out the strengths and limitations of SDM for CMC, There is so much we do not know about CMC and some healthcare professionals are more skilled at working with uncertainty than others. Thank you for the really beautifully written article on an important topic.
--	--

REVIEWER	Reviewer Name: Dr. Giles Birchley Institution and Country: University of Bristol Medical School, United Kingdom of Great Britain and Northern Ireland
REVIEW RETURNED	02-Feb-2023

GENERAL COMMENTS	I thought this was an interesting and informative review, that seems to have been well conducted. I had a couple of minor comments that I'd encourage you to consider addressing before it is published: Abstract- Line 6 - Typo "children with medical complexity is an increasing population" Discussion Page 12 Line 56 "Three studies showed poorer implementation in Black and Hispanic communities in the USA [35,36,38]. This is congruent with evidence showing that adults from minority ethnic or disadvantaged backgrounds experience more barriers to SDM [53]." – is the congruent evidence in [53] also from the USA ? I note the limitations indicate only US studies deal with race and that the impact may be different in other countries, but I was still unclear if [53] showed the US data accords more broadly with the data from other countries about the impact of ethnicity. Page 13 line 22-23 "Even when espousing SDM, HCPs often provide detailed information about their preferred treatment option first to convince or coerce the patient [7]" - I suggest that convincing and coercing are a little different – and it is not clear why convincing is not good practice, whereas it is clearer that coercing is not. Are both practices referred to in the source and why are they both deprecated? -Do you mean parent, rather than patient? (or is your source implying that the parent is the patient, not the child?) As I say, this are quite simple queries, but it would benefit the article if they were clarified/addressed.
---

VERSION 1 – AUTHOR RESPONSE

We would like to thank all three reviewers for their positive response and thoughtful comments. We have 'tracked changes' to highlight revisions.

Editor in Chief Comments to Author:

Abstract - Methods needs to state end date of search, ie papers published up to May 2022(?) and that only papers in English were considered

The above information was added to the abstract on page 2 - "Six databases were systematically searched for papers published in English up to May 2022:"

Table 3 Divide into four tables: one for qualitative primary research; one for quantitative and mixed methods primary research; one for opinion pieces; one for literature reviews. Table needs to have reference number of each paper.

Thank you for this suggestion, Table 3 has now been divided into 4 tables (pages 6-8) and reference numbers added. Presenting the included studies in this way makes it much easier to read.

Discussion add a sentence re the limitation of only including papers in English

Sentence added on page 12. "The review only included sources published in English which might have resulted in the exclusion of potentially valuable papers."

Reviewer: 1

Comments to the Author – Dr Diane Sellers

This scoping review presents a really helpful summary of the current literature subdivided in to different categories (qual research, quant research and opinion piece). It is well organised, clearly presented and comprehensive in its outlook.

Thank you for highlighting the importance of our work and this positive response.

The title suggests that the focus is on healthcare in a community setting although the search criteria and the papers you have included relate to both community and hospital based health care. E.g. Decisions about surgery are made with the surgeon; depending upon how services are organised, this may be with the family in acute hospital setting, hospital OP setting, or combined community clinic including surgeon providing outreach service. The configuration of decision making context and participants will influence the extent to which power imbalances are perceived and therefore changeable. I think you need to acknowledge that shared decision making in a community setting may not always be possible, especially when the key HCP involved in the decision is not present at the meeting. UK services are different across the country; North American Healthcare may be based strongly around hospitals with outreach in to the community. Differences in type of healthcare provided (insurance based, free healthcare with limitations, etc) will influence extent to which SDM can be considered. You may need to amend your title OR acknowledge the difficulty distinguishing between the two. Whilst I would support the notion that SDM is important in community healthcare, services will be configured differently to allow for different degrees of influence on critical healthcare decisions.

Thank you for this insightful comment highlighting the vast difference between different healthcare systems across the globe and how it influences hierarchy and decision-making. Dr Sellers commented that we need to consider either amending the title or acknowledging these issues. It was decided not to amend the title as it will change the essence of this review. Revisions were made instead to acknowledge this issue, a sentence was added in the discussion section on page 11 as well as an in the limitation section on page 12.

P11 - "All sources highlighted the importance of SDM; however, few studies explored the effectiveness of SDM for CMC, especially in community health settings. This might in part be due to the varying nature of service delivery models in different countries."

P12 - "Healthcare organisation and configuration vary across and within countries. It can range from mainly hospital-based services to services delivered in various hospital and community settings, services can be offered free at the point of contact or require payment by insurers or service-users. This variation could have impacted the identification of and comparability of studies."

The theme of "uncertainty" is a really key one and different health care professionals respond differently to "uncertainty" and this means decisions play out differently across different healthcare contexts. Because you have adopted headline QUAL review you have not explored themes underpinning each of the headings. Power hierarchies are rife within healthcare settings, especially in acute settings, with direct influence how decisions are made with families.

You point out the strengths and limitations of SDM for CMC, There is so much we do not know about CMC and some healthcare professionals are more skilled at working with uncertainty than others.

Thank you for the really beautifully written article on an important topic.

Thank you for these thoughtful comments, we are encouraged that this is important work in the field of shared decision-making for children with medical complexity and this further research is needed.

Reviewer: 2

Comments to the Author - Dr. Giles Birchley

I thought this was an interesting and informative review, that seems to have been well conducted. I had a couple of minor comments that I'd encourage you to consider addressing before it is published:

Thank you for this positive comment about our manuscript.

Abstract-

Line 6 - Typo "children with medical complexity is an increasing population"

We have reviewed this line but was unable to find the typo

Discussion

Page 12 Line 56 "Three studies showed poorer implementation in Black and Hispanic communities in the USA [35,36,38]. This is congruent with evidence showing that adults from minority ethnic or disadvantaged backgrounds experience more barriers to SDM [53]." – is the congruent evidence in [53] also from the USA ? I note the limitations indicate only US studies deal with race and that the impact may be different in other countries, but I was still unclear if [53] showed the US data accords more broadly with the data from other countries about the impact of ethnicity.

Thank you for your highlighting that the information presented was not clear to you as a reader and reviewer. We have reviewed the sentence and the source [53] and included a phrase on page 12 to highlight that the systematic review referred to included studies from 15 different countries. - "This is congruent with evidence from a systematic review that included studies from 15 countries, showing that adults from minority ethnic or disadvantaged backgrounds experience more barriers to SDM [53]."

Page 13 line 22-23 "Even when espousing SDM, HCPs often provide detailed information about their preferred treatment option first to convince or coerce the patient [7]"

- I suggest that convincing and coercing are a little different – and it is not clear why convincing is not good practice, whereas it is clearer that coercing is not. Are both practices referred to in the source and why are they both deprecated?

-Do you mean parent, rather than patient? (or is your source implying that the parent is the patient, not the child?)

Thank you for your thoughtful comment, we agree that convincing and coercing can have different meaning in the clinical context and that it is not useful to use them together in this context. After reviewing the source [7], we have rephrased the sentence to more clearly present the information from the original author. "Even when espousing SDM, HCPs often provide biased information in order to achieve a specific decisional outcome for instance by only providing information about their preferred treatment option [7]."

As I say, this are quite simple queries, but it would benefit the article if they were clarified/addressed.

Thank you for this positive comment about our manuscript.